# Water, Sanitation and Hygiene (WASH) in Schools in Low-Income Countries: A Review of Evidence of Impact

**DOI:** 10.3390/ijerph16030359

**Published:** 2019-01-28

**Authors:** Celia McMichael

**Affiliations:** School of Geography, The University of Melbourne, Carlton 3053, Australia; celia.mcmichael@unimelb.edu.au; Tel.: +61-3-8344-6704

**Keywords:** water, sanitation, hygiene, WASH, schools, intervention

## Abstract

Many schools in low-income countries have inadequate access to water facilities, sanitation and hygiene promotion. A systematic review of literature was carried out that aimed to identify and analyse the impact of water, sanitation and hygiene interventions (WASH) in schools in low-income countries. Published peer reviewed literature was systematically screened during March to June 2018 using the databases PubMed, Embase, Web of Science, the Cochrane Library, Science Direct, and Google Scholar. There were no publication date restrictions. Thirty-eight peer reviewed papers were identified that met the inclusion criteria. The papers were analysed in groups, based on four categories of reported outcomes: (i) reduction of diarrhoeal disease and other hygiene-related diseases in school students; (ii) improved WASH knowledge, attitudes and hygiene behaviours among students; (iii) reduced disease burden and improved hygiene behaviours in students’ households and communities; (iv) improved student enrolment and attendance. The typically unmeasured and unreported ‘output’ and/or ‘exposure’ of program fidelity and adherence was also examined. Several studies provide evidence of positive disease-related outcomes among students, yet other assessments did not find statistically significant differences in health or indicated that outcomes are dependent on the nature and context of interventions. Thirteen studies provide evidence of changes in WASH knowledge, attitudes and behaviours, such as hand-washing with soap. Further research is required to understand whether and how school-based WASH interventions might improve hygiene habits and health among wider family and community members. Evidence of the impact of school-based WASH programs in reducing student absence from school was mixed. Ensuring access to safe and sufficient water and sanitation and hygiene promotion in schools has great potential to improve health and education and to contribute to inclusion and equity, yet delivering school-based WASH intervention does not guarantee good outcomes. While further rigorous research will be of value, political will and effective interventions with high program fidelity are also key.

## 1. Introduction

Schools with adequate water, sanitation and hygiene (WASH) facilities have: a reliable water system that provides safe and sufficient water, especially for hand-washing and drinking; sufficient number of toilets for students and teachers that are private, safe, clean, and culturally and gender appropriate; water-use and hand-washing facilities, including some close to toilets; and sustained hygiene promotion [1]. Facilities should cater to all, including small children, girls of menstruation age, and children with disabilities. WASH conditions in schools in many low-income countries, however, are inadequate with associated detrimental effects on health and school attendance [2]. An evaluation by UNICEF [3] found that in schools in low-income countries, only 51% of schools had access to adequate water sources and only 45% had adequate sanitation. 

Globally, school-based WASH interventions variously aim to: (i) reduce the incidence of diarrhoea and other hygiene related diseases; (ii) improve school enrolment, school performance, and attendance; and (iii) influence hygiene practices of parents and siblings whereby children act as agents of change in their households and communities. However, evidence assessing the impact of school-based WASH interventions has been mixed. Two previous reviews of studies of the impact of school-based WASH interventions have shown mixed results on outcome measures such as knowledge, attitudes and practices, school attendance, and health [2,4]. The review by Jasper et al. [2] had a global focus and most included studies (*n* = 41) were from high- and middle-income countries (e.g., United States, United Kingdom); Joshi and Amadi [4] also had a global focus including studies from North America and Europe and their review was confined to studies (*n* = 15) published between 2009–2012. 

The objective of this review is to analyse published peer-reviewed journal articles that focus on WASH in schools in low-income countries. The review focuses on intervention-based studies and key outcome measures including: health among school students (e.g., diarrhoeal disease and other hygiene-related diseases); WASH knowledge, attitudes and hygiene behaviours among students; changes in disease burden and hygiene behaviours in students’ households and communities; changes in student enrolment and school attendance. The review also considers the under-reported indicator of intervention fidelity. The review highlights gaps in knowledge and potential future research directions.

## 2. Materials and Methods 

Published peer reviewed journal articles were included that examined the impacts of school-based WASH intervention in low-income countries. WASH interventions included: hand-washing initiatives (e.g., water, wash basins, soap, drying devices); drinking water initiatives; improved sanitation (improved toilets, facilities for menstruation); and hygiene behaviour initiatives (e.g., handwashing with soap, hygiene education). Reported outcomes include: educational outcomes (i.e., school attendance, school dropout); hygiene behaviours, knowledge and attitudes; and health (i.e., WASH-related illness). Intervention fidelity—adherence to intervention delivery standards—was also reported in several studies (either as an ‘exposure’ or ‘outcome’). Article inclusion was restricted to those with a focus on low-income countries, defined as countries with a Gross National Income (GNI) per capita (calculated using the World Bank Atlas method) of 1005 USD or less in 2016. The review was restricted to articles for which the abstract and article was available in English language.

Descriptive studies of school-based WASH conditions, without evaluative focus on intervention impacts, were excluded [5,6]. Morgan et al. [5], for example conducted a cross-sectional survey of 2270 WASH intervention beneficiary schools in Ethiopia, Kenya, Mozambique, Rwanda, Uganda and Zambia and found that fewer than 23% of rural schools met World Health Organization recommended student-to-latrine ratios. While descriptive studies provide important insight into the context and challenges for WASH in schools, they are not the focus here. 

The following electronic databases were searched during March to June 2018: PubMed, Embase, Web of Science, the Cochrane Library, Science Direct, and Google Scholar. The search was based on the keywords: WASH or water or sanitation or soap or hygiene or “hand hygiene” or “hand wash*” AND school or attendance AND “low income” or “developing country” or “developing nations”. For example, in Embase the following search terms were deployed: (WASH OR water OR hygiene OR “hand hygiene” OR “hand wash*” OR sanitation OR Soap* OR “child* health”) AND (school OR attendance) AND (“low income country” OR “developing country”). References of included articles were systematically searched for relevant documents. There were no publication date restrictions.

## 3. Results

### 3.1. Systematic Review and Yielded Studies

The initial search terms identified 1498 publications; 11 additional articles were identified from other sources. The secondary screening—based on the title—identified 119 articles with a potential focus on WASH in schools in low-income countries. Thirty eight of these articles met the inclusion criteria, following screening by abstract and then full text. Bibliographies of these references identified no additional articles (see Figure 1).

For each article, a summary of key information was tabled: i.e., country of study, study design, study population (number of schools, children, and/or their age), exposure/intervention, outcome measure, key findings. As the studies use diverse methods and outcome measures no attempt was made to weight the value of findings according to study quality, or to conduct meta-analysis of study findings. Of the 38 articles: 47% reported the intervention impact on diarrhoeal disease and other hygiene-related diseases in school students; 34% reported changes in WASH knowledge, attitudes and hygiene behaviours among students; 16% reported impact on disease burden and hygiene behaviours in students’ households and communities; 32% reported changes in student enrolment and school attendance; and 11% reported on intervention fidelity (see Table 1). Twelve studies reported outcome measures across more than one category [7,8,9,10,11,12,13,14,15,16,17,18] (see Table 1). 

Countries of focus included Bangladesh, Burkina Faso, Cambodia, China, Colombia, Egypt, Ethiopia, Ghana, India, Indonesia, Kenya, Lao People’s Democratic Republic, Mali, Niger, Nepal and Tanzania. Study methods included cross-sectional survey, non-randomized trial, cluster-randomized trial, and before and after intervention studies. Study design is identified in Table 2.

### 3.2. Reduced Diarrhoea and other WASH-Related Diseases in School Students

Despite the biological plausibility that improvements in school WASH conditions will be beneficial for pupil health, results from school-based WASH evaluations have been mixed. There is evidence that WASH in Schools programs have a positive impact on child health, including reductions in diarrhoeal disease and other hygiene-related diseases. Migele et al. [28] examined the impact of a simple school-based water treatment and hand-washing intervention in a boarding school in Kenya: i.e., clay pots modified with narrow mouths and ceramic lids, taps for drinking water, plastic tanks with taps for hand washing, WaterGuard (i.e., sodium hypochlorite solution) for drinking water, and soap for hand washing. Before-and-after rates of diarrhoea disease (with no control schools) indicated a more than 50% reduction in recorded cases of diarrhoea among students. In their evaluation of WinS interventions in Mali, Trinies et al. [18] found that, as compared with control schools, there were lower odds of students in beneficiary schools reporting diarrhea (OR 0.71, 95% CI 0.60–0.85) or respiratory infection symptoms (OR 0.75, 95% CI 0.65–0.86) in the past week. And a study in rural Kenya [15] found that school-based water treatment and hygiene programs resulted in a decrease in rates of acute respiratory illness, although no decrease in acute diarrhea was observed. Improving school-based WASH can also reduce other hygiene-related diseases, such as soil-transmitted helminth (STH) infection [7,21,22]. For example, Bieri et al. [7] found that among Chinese school-children, the incidence of infection with STHs was 50% lower in the intervention group that received a STH education package than in the control group (4.1% vs. 8.4%, *p* < 0.001). And in Mali, Freeman et al. [22] found that provision of school-based sanitation, water quality, and hygiene improvements reduced reinfection of some STHs after school-based deworming, but the magnitude of the effects were helminth species-specific.

Results, however, are not uniformly clear or positive. In an evaluation of a hand-washing promotion program in Chinese primary schools, rates of diarrhoea were too low in both intervention and control groups to identify attributable differences in prevalence [35]. Some studies indicated that basic interventions that include hygiene promotion, water treatment, and behaviour change did not reduce rates of diarrhoeal disease [11,15]. In a multi-country study, Dujister et al. [20] found that the STH prevalence at baseline and at follow-up did not significantly differ between intervention schools (that provided deworming and improved handwashing) and control schools. And a study by Greene et al. [25] conducted in schools in western Kenya found that hygiene promotion and water treatment did not reduce risk of *Escherichia coli* presence on pupils’ hands; further, the addition of new latrines to intervention schools significantly increased *E. coli* presence among girls (RR = 2.63, 95% CI 1.29–5.34) which they attributed to an absence of sufficient hygiene behaviour change, and lack of soap, water, and anal cleansing materials. It is important to note, however, that presence of *E. coli* on hands is a variable that is difficult to interpret in terms of disease risk and outcomes. 

Context is important. For example, Freeman et al. [11] found that local water availability affected the impact of school-based WASH interventions on diarrhoea rates among pupils. Pupils attending ‘water-scarce’ schools (in which there was no dry-season water source within 1km) that received WASH intervention (including water-supply improvement, hygiene promotion and water treatment, and sanitation improvements) reported a reduction in diarrhoea incidence and days of illness; they reported a 56% difference in the risk of diarrhoea for pupils attending intervention vs. control schools in water-scarce sites (adjusted risk ratio (aRR) 0.34, 95% CI 0.17–0.64). No statistically significant effect was detected for any intervention in ‘water-available schools’, nor for ‘water-scarce’ schools that received only hygiene promotion and water treatment. Similarly, Garn et al. [44] found that in water-scarce schools in Kenya, there was reduced prevalence of diarrhea among pupils attending schools that adhered to two or three intervention components (prevalence ratio 0.28, 95% CI 0.10–0.75), compared with schools that adhered to zero components or one. It was not clear why results were different in water-scarce versus water-available schools, but it is possible that WASH interventions in water-scarce schools were more comprehensive. 

There is widespread recognition that WASH infrastructure and resources are important foundations for hygiene behaviour change and reduced risk of WASH-related diseases. There is evidence, however, that latrine construction, without other supporting water and hygiene-related interventions, is not effective at reducing diarrhoeal disease [11, 20). Possible explanations are that without broader hygiene promotion and latrine maintenance efforts, construction of latrines alone may not result in their use or (conversely) latrines may increase exposure to faecal pathogens if they are poorly maintained, used incorrectly, or if hygiene resources are not available during and after use [11,36]. The health benefits of improved WASH infrastructure and resources in schools may depend on consistent availability of soap and water for handwashing and on conditions of the latrines, not only pupil to latrine ratios [26]. 

### 3.3. Improved WASH Knowledge, Attitudes and Hygiene Behaviours

Thirteen studies measured WASH knowledge, attitudes and hygiene behaviours among students (see Table 1); all found evidence of improved knowledge, attitudes and behaviours associated with WinS program. Dreibelbis et al. [29] report findings of an intervention that aimed to improve hand-washing after toilet use among students in two primary schools in rural Bangladesh. Dedicated locations for hand-washing were constructed in both schools. Two nudges were implemented: first, connecting latrines to hand-washing stations via brightly painted paved pathways; second, painting footprints on pathways guiding students to the handwashing stations and handprints on stations. Soap was provided and schools were asked to make soap available and refill water storage containers each day. At baseline, hand-washing with soap (HWWS) was low (4%); this increased to 68% the day after nudges were completed and 74% at both 2 weeks and 6 weeks post intervention. The high rates of observed handwashing post-intervention suggest that nudges can have sustained effects on hygiene behaviours. A related cluster-randomized trial in schools Bangladesh [30] demonstrated comparable increases in rates of handwashing with soap five months after intervention both for a nudge intervention (paved path with painted shoe-prints and arrows connecting latrines to the handwashing facility, painted handwashing station with handprints and a dedicated location for soap) and high intensity hygiene education initiatives. La Con et al. [31] found that installation of water and handwashing stations in schools in rural Kenya, coupled with WASH education, enabled student handwashing with stations located closer to latrines (<10 m) used much more frequently. One randomized cluster trial in rural Kenya [17] examined the impact of provision of regular soap and latrine cleaning materials and hygiene education; pupil hand-washing rates following toileting were observed to be 32–38% in intervention schools compared to 2% of students in control schools. Another randomized cluster trial in urban Nairobi, Kenya, examined the impact of teacher hygiene training and provision of regular alcohol-based hand sanitizer or liquid soap; pupil hand-washing rates following toileting were observed to be 82% at schools with sanitizer, 38% at schools with soap, and 37% at control schools [16]. 

### 3.4. Reduced Disease Burden and Improved Hygiene Practices in Households and Communities

In addition to limiting pathogen transmission in the public domain—such as at schools—school-level WASH interventions may also reduce community disease burden and improve hygiene knowledge. One study in Kenya found that in water-scarce areas, school-based WASH interventions that included improvement in water supply reduced diarrhoea among school students’ siblings under the age of five who were not attending school [33]. The authors suggest this could be due to diffusion of improved hygiene practices and behaviours in both home environments and community, or interruption of pathogen transmission in school contexts thereby reducing exposure and transmission in domestic environments. Another study in Kenya documented transfer of knowledge from school students to their parents, identifying increased parental awareness and household use of water treatment with flocculent disinfectant following student hygiene education and provision of water treatment products to students; improved household water treatment practices were sustained over one year [32]. A study of a school-based WASH intervention in Kenya documented the transfer of knowledge about point-of-use water treatment practices and increased utilisation of WaterGuard in student’s households as indicated by having chlorine residuals in stored water; parents also reported improved hand-washing and 38% of parents demonstrated correct hand-washing technique [14]. However, based on their study in Burkina Faso, Erismann et al. [10] warn that although children can promote health messages to family members, effective behaviour changes among family members is more difficult to achieve due to the challenge of changing practices and the broader constraints that limit improved behaviours (e.g., water scarcity).

### 3.5. Improved Student Enrolment and Attendance

In this review, twelve studies in low-income countries were identified that examined the impact of school-based WASH programs on student absence and enrolment. Improved school WASH conditions may reduce student absence by providing services (including, importantly, for girls who are menstruating) and by reducing illness transmission [45]. There is some evidence that improved hand-washing with soap at school can reduce illness in school-aged children thereby reducing absence from school [11,14,15,18,21,35,41]. 

Interventions that deliver hand-washing promotion and point-of-use water treatment have reported reductions in student absence of between 21% [32] and 61% [38] with one study specifically identifying reduced absence among girls (i.e., 58% reduction in the odds of absence for girls) [21]. A school-based water and hygiene intervention in public primary schools in Kenya found a decrease in student absence of 35% relative to baseline as compared to a 5% increase in neighbouring schools [14]. Talaat et al. [41] identified a 21% reduction in school absence from all illnesses (e.g., diarrhea, conjunctivitis, influenza) as a result of an intensive hand-washing campaign in Egypt; absences caused by influenza-like illness, diarrhea, conjunctivitis, and laboratory-confirmed influenza were reduced by 40%, 33%, 67%, and 50%, respectively. A small pilot study in Ghana entailed provision of sanitary pads and puberty education to adolescent girls in both intervention and control schools, with the intervention found to significantly improve attendance [39]. Evaluation of a comprehensive WASH intervention in schools in Bangladesh—using a non-experimental survey design—reported a 9–12% reduction in school absence among girls (varying between schools) [42]. A trial of school-based WASH interventions in Kenya found that cleanliness of latrines was strongly correlated with recent student absence [37]. And a study of hand-washing intervention in Chinese primary schools found that the expanded intervention (standard government education plus hand-washing program, soap for sinks, and peer hygiene monitors) reported 42% fewer absence episodes and 54% fewer days of absence, and the standard intervention (handwashing program) reported 44% fewer absence episodes and 27% fewer days of absence [35]. 

Some intervention studies, however, found no evidence of impact on attendance. A study in the Chitwan region of Nepal [40] trialled the use of menstrual cups (a silicone cup used internally for menstrual flow management) with a small sample of schoolgirls. The study found the technology had no impact on school attendance or school test outcomes; the authors suggest this is because the technology assisted only with management of blood, and did not reduce cramps which were reported as the primary reason for non-attendance. However, the study had several limitations including self-reporting of menstrual cup usage, and lack of consideration of existing water and sanitation facilities in schools. And a trial in Kenya to assess the impact of a scalable, low-cost, school-level latrine cleaning intervention on pupil absence did not find a reduction in absenteeism; the authors hypothesised that the additional impact of cleaning may not have been sufficient to reduce absence beyond reductions attributable to the original WASH intervention [36]. 

### 3.6. Intervention Fidelity

Effectiveness of interventions is associated with the typically unmeasured and unreported ‘output’ and/or ‘exposure’ of intervention delivery including program fidelity and adherence. Three studies reported on intervention fidelity but did not draw conclusions as to its effect on measured outcomes. Chard and Freeman [9] report on a WASH intervention in Laotian primary schools and found inadequate school-level adherence to project outputs (e.g., soap provision, water availability, hygiene promotion activities); the differential impact of school-level intervention fidelity on measured hygiene behaviours (e.g., toilet use and daily hygiene activities) was not reported. Alexander et al. [43] assessed whether student and parental monitoring and additional funding for repairs and maintenance affected the fidelity and effectiveness of school-based WASH service provision in 70 schools in Western Kenya; no clear results emerged. Hetherington et al. [12] reported on an initiative in Tanzania that aimed to engage high-school students and the wider community in improving sanitation and hygiene. While they noted challenges of intervention adherence and fidelity—including timing of activities, communication between schools and local coordination, and inadequate supplies and allowances to support activities—the impact of these challenges on the primary outcome measures (i.e., hygiene knowledge, attitudes, behaviours) was not assessed. Garn et al. [44] provide rare evidence of the impact of intervention adherence and found that among water-scarce schools in Kenya improved adherence resulted in reduced prevalence of diarrhoea among pupils.

## 4. Discussion

Access to WASH facilities and hygiene behaviour change education in schools contribute to inclusion, dignity, and equity. From a human rights perspective, WASH in schools is considered essential. The Sustainable Development Goals (SDGs) implicitly highlight the need to expand WASH beyond household settings, in the effort to achieve universal and equitable access to safe and affordable drinking water, sanitation and hygiene for all. The SDGs explicitly refer to WASH in Schools in Target 4.a via the indicator of the “proportion of schools with access to: (e) basic drinking water; (f) single-sex basic sanitation; and (g) basic handwashing facilities” [46]. However, the aim is to not only provide adequate ratios, but to ensure positive outcomes across diverse measures including diarrhoeal disease and other WASH-related diseases, hygiene behaviour and school attendance. 

There is biological plausibility supporting the health and educational benefits of providing WASH in schools, as well as rights-based arguments for WASH in Schools. The studies in this review indicate that school-based WASH interventions can protect against diarrhoea and other WASH-related illness such as soil-transmitted helminths and acute respiratory infections, increase WASH-related knowledge and practices, and improve educational outcomes including reduced absence. 

Fourteen (78%) of the 18 publications that reported disease-related outcomes found reductions in diarrhoeal disease and other hygiene-related diseases, such as respiratory illness and soil-transmitted helminths, among students at intervention schools (c.f. [7,18,21,28]). Of these publications reporting positive health outcomes, however, more than half also reported that there were no statistically significant reductions for some disease-related outcomes: e.g., intestinal parasitic infections prevalence, but not undernutrition, was found to decrease [10]. Four of the 18 publications reported no evidence of reduced risk for the primary disease-related outcome measures, including soil-transmitted helminths and *E. coli* on pupils’ hands [17,20,25,26]. 

All of the 13 publications that examined changes in WASH knowledge, attitudes and hygiene behaviours reported evidence of positive change among students in intervention schools including hand-washing with soap or sanitizer [8,16,29,30,31], improved knowledge of WASH-related diseases, and improved hygiene habits [7,13]. 

Six studied examined whether WASH interventions in schools led to reductions in the family and community burden of WASH-related diseases and improved WASH knowledge at the family and community level. They provide very limited evidence of improvements in WASH-related knowledge and behavior and reduced WASH-related disease among family [14,32,33]. Further research is required to understand whether and how school-based WASH interventions can improve hygiene habits and disease-related outcomes among wider family and community members [29]. 

Demographic factors are key predictors of student absence from school, including gender and socio-economic status [37]. Nonetheless, WASH-related illnesses have been estimated to result in hundreds of millions of days of school absence [47]. Twelve publications examined the impact of school-based WASH interventions on student absence in low-income countries and the findings were mixed. There is some evidence that improved hand-washing with soap at school, provision of sanitary pads, maintained and clean latrines can reduce absence in school-aged children (c.f. [11,18,35,37,42]), but a few studies found that school-based WASH interventions had no impact on student attendance [36,40]. 

Importantly, intervention effectiveness is affected by intervention delivery, including program fidelity and adherence. Freeman et al. [11] warn that suboptimal intervention fidelity often means that researchers evaluate the effectiveness of interventions in real-world settings, not ideal ‘best practice’ for WASH environments. Yet, while various publications mention the challenges of fidelity and adherence in school-based WASH interventions, their impact on outcomes is rarely assessed; only one study in schools in Kenya specifically demonstrated that improved intervention adherence resulted in reduced prevalence of diarrhoea among pupils [44]. Studies such as these highlight that ensuring consistent and effective delivery of WASH interventions in low-resources contexts, including school-based interventions, remains a challenge.

So, there is no universal blueprint and effects are not consistent between studies as both context and intervention type matter. For example, the effectiveness of an intervention in reducing diarrhoeal disease may be based on background rates of disease, pathogen-pathways in specific environments, student populations, baseline WASH conditions such as water availability, and broader social, political and economic contexts [11,44]. Several publications emphasise that combined interventions that include multiple components—for example, latrine construction, hygiene promotion, latrine maintenance, and sustained provision of resources such as soap and water for handwashing—are more effective at reducing WASH-related diseases than single interventions such as construction of latrines [11,21,36].

Evaluative research of WASH in Schools encounters challenges which influence results and their interpretations, including: restrictions in randomisation, the potential of crossover effects, and circumstances beyond the researchers’ control such as the interference of other health programmes. The definition of illness outcomes such as “diarrhea” are not uniform across studies which makes inter-study comparison difficult. And, importantly, evaluations of WASH interventions in low-resource settings often measure outcomes—such as diarrheal disease—via self-report, an approach prone to recall and social desirability biases, subjective interpretations of the definition of “diarrhea”, and imprecise measurements of incidence [9]. It is notable that of the 18 studies in this review that report disease-related outcomes, ten (56%) included objective rather than self-reported measures of disease and infection: for example, fecal samples were examined for soil-transmitted helminths, intestinal protozoa and other parasites [7,8,10,11,20,22,24,26], blood samples were collected to measure blood hemoglobin concentration [26], and hand-rinse samples were analysed for *E. coli* [17,25].

The theory of change embedded in project design also influences the nature of an intervention and its delivery. In their evaluation of Project SHINE (Sanitation and Hygiene INnovation in Education) in Tanzania, for example, Hetherington et al. [12] highlighted the value of strategies that enable communities to develop locally sustainable approaches to improving their health, in contrast to other models (e.g., Community Led Total Sanitation) which incorporate shaming and disgust techniques to promote behaviour change. Theories of change must be considered to fully understand effectiveness, or lack thereof, rather than reducing interventions to processual elements of exposure and outcome. 

Notably, several studies have examined the onset and management of menses in low-income countries, with a specific focus on the challenges of menstrual hygiene management (MHM) in school environments (e.g., negative attitudes, limited health and sexuality information, inadequate facilities and privacy) (c.f. [48,49,50]). However, these studies are qualitative and/or descriptive; very few intervention studies include a focus on menstrual hygiene management in schools in low-income countries [39,40,43].

This review contributes to understanding of the impact of school-based WASH interventions beyond the two existing reviews of school-based WASH by Jasper et al. [2] and Joshi and Amadi [4]. First, these two existing reviews have no restrictions on study location and more than two thirds of the 41 articles in the review by Jasper et al. [2] report findings of research conducted in high-income countries (e.g., United Kingdom, United States, Germany) and almost one third of the 15 articles in the review by Joshi and Amadi [4] were conducted in developed countries; this review has an explicit focus on low-income countries where there is the greatest need for improved access to safe drinking water, improved sanitation, handwashing facilities and hygiene education [47]. Second, these reviews necessarily include only publications available up to 2012: the Jasper et al. review [2] had no time restriction on the date of publication and the search was conducted in 2010 and updated in 2012; the Joshi and Amadi [4] review was restricted to studies published between 2009 and 2012 and the search was conducted in 2013. In this review, however, twenty-five of the 38 studies included were published after 2012. The contribution of this review, then, is its explicit focus on low-income countries and its inclusion of the substantial body of relevant research published in the last several years.

## 5. Conclusions

It is important to better understand disease-related and educational outcomes of school-based WASH interventions. This can help governments and donors allocate resources to school-based WASH interventions and enable agencies to design and implement effective interventions [11]. Intervention studies of WASH in schools in low-income settings are both expensive and challenging. There is, arguably, no need for additional large-scale epidemiological studies on the impact of WinS on diarrhoea among students as numerous studies have found evidence of positive outcomes related to diarrhoeal disease [11]. There is, however, still a need to better understand the differential impacts of different types of WinS programmes for broader health and educational outcomes, the extent to which students operate as change agents in wider communities, the role of independent variables including gender and socio-economic status, and the effect of targeted initiatives on menstrual hygiene management and girls’ school attendance. Further, there is value in conducting process evaluations that identify opportunities and challenges within program implementation, including theories of change and intervention fidelity. Political will and financing and effective delivery of interventions will be required to ensure universal access to WASH in Schools including in low-income countries.

## Figures and Tables

**Figure 1 ijerph-16-00359-f001:**
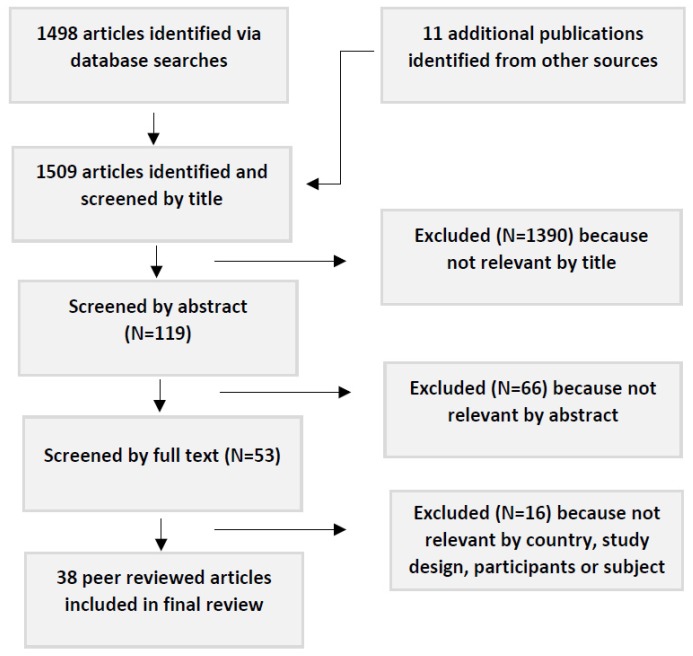
Flow chart showing procedure for article selection.

**Table 1 ijerph-16-00359-t001:** Outcome measures reported in included articles (*n* = 38).

Outcome measure	% of studies	Studies	
Impact on diarrhoeal disease and other hygiene-related diseases in school students	47% (18/38)	Bieri et al. 2013 [7] Boubacar Maïnassara & Tohon 2014 [8] Chard et al. 2018 [19] Dujister et al. 2017 [20] Erismann et al. 2017 [10] Freeman et al. 2012 [21] Freeman et al. 2013 [22] Freeman et al. 2014 [11] Freeman et al. 2015 [23]	Garn et al. 2016 [24] Greene et al. 2012 [25] Grimes et al. 2017 [26] Koopman 1978 [27] Migele et al. 2007 [28] Patel et al. 2012 [15] Pickering et al. 2013 [16] Saboori et al. 2013 [17] Trinies et al. 2016 [18]
Changes in WASH knowledge, attitudes and hygiene behaviours among students	34% (13/38)	Bieri et al. 2013 [7] Boubacar Maïnassara & Tohon 2014 [8] Chard & Freeman 2018 [9] Dreibelbis et al. 2016 [29] Erismann et al. 2017 [10] Grover et al. 2018 [30] Karon et al. 2016 [13]	Hetherington et al. 2017 [12] La Con et al. 2017 [31] O’Reilly et al. 2008 [14] Patel et al. 2012 [15] Pickering et al. 2013 [16] Saboori et al. 2013 [17]
Impact on disease burden and hygiene in students’ households/communities	16% (6/38)	Blanton et al. 2010 [32] Dreibelbis et al. 2014 [33] Erismann et al. 2017 [10]	Freeman & Clasen 2011 [34] Karon et al. 2016 [13] O’Reilly et al. 2008 [14]
Changes in student enrolment and school attendance	32% (12/38)	Boubacar Maïnassara & Tohon 2014 [8] Bowen et al. 2007 [35] Caruso et al. 2014 [36] Dreibelbis et al. 2013 [37] Freeman et al. 2014 [11] Hunter et al. 2014 [38]	Montgomery et al. 2012 [39] O’Reilly et al. 2008 [14] Oster & Thornton 2009 [40] Talaat et al. 2011 [41] Trinies et al. 2016 [18] UNICEF 1994 [42]
Intervention fidelity	11% (4/38)	Alexander et al. 2013 [43] Chard & Freeman 2018 [9]	Garn et al. 2017 [44] Hetherington et al. 2017 [12]

**Table 2 ijerph-16-00359-t002:** Published evaluations of WASH in schools in low-income countries.

Number	Authors	Country	Study Design	Sample	Exposures/Intervention	Measured Outcomes	Key Findings
1	Alexander et al. 2013 [43]	Kenya	Cluster-randomized trial	70 schools divided into a control group (*n* = 25) and three intervention groups (*n* = 15 per group)	Intervention schools received a budget for WASH-related items. One group received no further intervention. Second group received funding for WASH attendant and WASH infrastructure repairs. Third group given guide for monitoring WASH conditions.	Quality of school latrines, rainwater-harvesting systems, handwashing facilities, and other school infrastructure; maintenance and cleanliness of latrines; drinking water treatment. Intervention fidelity.	Intervention schools made significant improvements in provision of soap, handwashing water, treated drinking water, and clean latrines. Unclear whether expanded interventions out-performed budget-only intervention.
2	Bieri et al. 2013 [7]	China	Cluster-randomized trial	38 schools	Schools randomly assigned to a health-education package—a cartoon video about STHs, pamphlet, teacher-training workshop, essay competition—or a control package of a health-education poster.	Infection rates with soil-transmitted helminths, knowledge about soil-transmitted helminths, self-reported hygiene behaviours, and observed hand-washing behaviour.	Health-education package increased students’ knowledge of STHs, improved hygiene behaviour, and reduced STH infection by 50% within 1 school year.
3	Blanton et al. 2010 [32]	Kenya	Before and after intervention study	17 schools (666 students at baseline)	Installation of drinking water and hand-washing stations in schools; teacher training on WASH promotion; hygiene education for students; distribution of instructional comic books to students; school children encouraged to promote water treatment and handwashing in schools and households.	Water handling survey of pupils’ parents at 3 and 13 months. Household stored water tested for chlorine at 3 and 13 months.	The program resulted in pupil-to-parent knowledge transfer around water treatment and increases in household water treatment practices that were sustained over 1 year and reduction in student absentee rates.
4	Boubacar Maïnassara and Tohon 2014 [8]	Niger	Before and after intervention study	6 schools (sample of children aged 7 to 12 years; *n* = 720)	Installation of clean water outlets, latrines, handwashing stations and clean drinking-water; student, teacher and parent hygiene education; display of hygiene promotion materials.	Student-reported symptoms of diarrhoea, water consumption habits, sources of drinking water at school, latrine usage, hygiene behaviours. Teacher-reported student absence. STH infection diagnosed via stool samples.	A reduction in self-reported diarrhoea cases and abdominal pain was noted in both intervention and control schools. Student absence increased post-project, but not as much as in control schools. Carriage of at least one parasite reduced in intervention schools, but findings were not statistically significant. There was an increase in reported handwashing in intervention schools.
5	Bowen et al. 2007 [35]	China	Cluster randomized trial	87 primary schools	Control: standard government hygiene education (i.e., annual statement about washing hands after using the toilet and before eating). Standard intervention: standard govt. education plus handwashing program. Expanded intervention: standard govt. education plus handwashing program, soap, and peer hygiene monitors.	Student absence rates	Provision of standard and expanded hand-washing promotion program and soap in schools was associated with significantly reduced days and episodes of student absence.
6	Caruso et al. 2014 [36]	Kenya	Cluster-randomized trial	17,564 pupils in 60 schools	Low-cost environmental-level latrine cleaning intervention as an added element following previously received WASH improvements in schools.	Latrine conditions and use; student absence	The addition of a latrine cleaning component may not have affected student absence beyond reductions attributable to the original school-based intervention.
7	Chard and Freeman 2018 [9]	Laos People’s Democratic Republic	Randomized controlled trial	100 public primary schools: 50 intervention and 50 comparison.	Interventions schools. Hardware; water supply, school sanitation facilities, handwashing facilities. Software; classroom ceramic water filter, group handwashing with soap at critical times, student-led cleaning and maintenance of toilets, school compound maintenance.	WASH behaviors, i.e., student toilet use, daily group handwashing, individual handwashing practice. Intervention fidelity.	Intervention schools had sustained service improvements: i.e., access to toilets, handwashing facilities, and safe drinking water. There were improvements in pupils’ WASH behaviors: use of school toilet, increased handwashing with soap, and habitual daily group handwashing. 88% of schools received the intervention as per design; school-level adherence was lower.
8	Chard et al. 2018 [19]	Mali	Matched-control trial	42 primary schools; 21 intervention and 21 matched comparison Schools.	A comprehensive school-based WASH intervention: school WASH infrastructure, WASH supplies and hygiene kits, behaviour change and training activities for students and teachers and within wider community, establishment of school-level financial, governance and management systems.	Vector-transmitted disease (dengue), food/water transmitted enteric disease (*Escherichia coli* and *Vibrio cholerae*), and person-to-person transmitted enteric disease (norovirus).	Food/water-transmitted enteric disease and person-to-person transmitted enteric disease was lower among pupils attending beneficiary schools. There was no evidence of difference in vector-transmitted disease.
9	Dreibelbis et al. 2013 [37]	Kenya	Cross-sectional survey	7966 children from 3857 households, enrolled in 175 primary schools.	Existing school WASH conditions. Household WASH conditions and knowledge, attitudes and practice. Household demographic characteristics.	Student school absence (household-reported)	School latrine cleanliness was the only school WASH factor associated with odds of absence. Demographic features (e.g., gender, SES, household characteristics) were important predictors of absence.
10	Dreibelbis et al. 2014 [33]	Kenya	Cluster-randomized trial	185 schools: ‘Water-available’ schools with water source within 1 km (*n* = 135); ‘Water-scarce’ schools with no water source within 1 km (*n* = 50).	Schools allocated to different interventions (including hygiene promotion + water treatment; hygiene promotion, water treatment, sanitation; control).	Prevalence of diarrhoea and two-week period prevalence of clinic visits among children <5 years with at least one sibling attending a program school.	In water-scarce areas, school WASH interventions that improve water supply can reduce diarrheal diseases among siblings of students.
11	Dreibelbis et al. 2016 [29]	Bangladesh	Before and after intervention study	2 primary schools (220 and 514 students)	Inexpensive nudges—i.e., environmental cues to prompt behaviour change—to encourage hand-washing with soap. Nudges included connecting latrines to handwashing station via brightly painted paved pathways; painting foot prints on pathways to guide students to handwashing stations.	Handwashing with soap (HWWS)	HWWS was increased the day after nudges were completed (from 4% to 68%) and further increased to 74% at two and six weeks post intervention. Nudge-based interventions have potential to improve HWWS among school children.
12	Dujister et al. 2017 [20]	Cambodia, Indonesia and Lao PDR	Non-randomized clustered controlled trial	1847 children attending public elementary schools at baseline; 1499 children at follow-up.	School-based “FIT programme” including daily group handwashing with soap and tooth-brushing with fluoride toothpaste, biannual school-based deworming, group handwashing facilities	Parasitological, weight, and oral health status of children.	The prevalence of STH infection, thinness, and oral health (odontogenic infection) did not significantly differ between baseline and follow-up, nor between intervention and control schools. Dental caries were significantly reduced.
13	Erismann et al. 2017 [10]	Burkina Faso	Cluster-randomized trial	360 randomly selected children, aged 8–15 years,	The intervention included school garden, nutrition, and WASH components. The WASH component involved installation of latrines and handwashing stations, rehabilitation of water pumps, and safe drinking water stations in classrooms. Hygiene and nutrition education was provided to teachers, school directors and community representatives. Treatment was provided to children found to be anaemic or infected with intestinal parasites.	Prevalence of intestinal parasitic infection and nutritional status. Children’s health knowledge, attitudes, and practices. *Escherichia coli*-positive in drinking water samples from student’s households.	At end-line, the prevalence of intestinal parasitic infections decreased significantly in the intervention schools compared to control schools. Indices of undernutrition did not decrease in intervention schools. Safe handwashing practices significantly improved in the intervention schools.
14	Freeman and Clasen 2011 [34]	India	Randomized case-control intervention study	56 primary schools and 16 middle schools	Classrooms provided with a commercial water purifier; basic hygiene and water treatment information provided to students, parents, and teachers.	Awareness and uptake of effective water treatment practices at home.	No evidence that school-based intervention led to increased awareness or adoption of improved water management practices in homes. Membership in self-help group associated with uptake of water purifier.
15	Freeman et al. 2012 [21]	Kenya	Cluster-randomized trial	185 schools, including: 135 water-available schools (water source within 1 km) and 50 water-scarce schools (no water source within 1 km)	In water-available and water-scare sites, schools randomly allocated to different interventions (including hygiene promotion + water treatment; hygiene promotion, water treatment, sanitation; control).	Period prevalence and days of diarrhoeal illness in pupils that received different WASH interventions (including control schools).	In the absence of adequate water supplies, school-based WASH—i.e., water-supply improvement, hygiene promotion and water treatment, improved sanitation - can reduce diarrhoea.
16	Freeman et al. 2013 [22]	Kenya	Cluster-randomized trial	40 government primary schools.	Schools randomly allocated: - Deworming plus a comprehensive school-based water treatment, sanitation, and hygiene intervention - School-based deworming only	Infection with soil-transmitted helminths: hookworms, roundworm, and whipworm. Secondary outcome included the prevalence and egg count of trematode *Schistosoma mansoni*.	The intervention reduced reinfection prevalence and egg count of roundworm, Ascaris lumbricoides. No evidence of effectiveness for *Trichuris trichiura*, hookworm, or *Schistosoma mansoni* reinfection.
17	Freeman et al. 2014 [11]	Kenya	Cluster-randomized trial	185 public primary schools: 135 were water-available schools (water source within 1 km); 50 were water-scarce schools (no water source within 1 km.	Schools randomly allocated: - Hygiene promotion and water treatment intervention including closed buckets, taps, drinking water storage, water disinfection; teacher training on hygiene behaviour change promotion. - As above plus ventilated improved pit (VIP) latrines. - Control group: receive intervention at conclusion of study.	Prevalence of diarrhoea, number of days of illness with diarrhoea, pupil absence.	Pupils attending ‘water-available’ schools that received hygiene promotion and water treatment (HP&WT) only, or WP&WT and sanitation improvements, showed no difference in illness compared to control schools. Pupils in ‘water-scarce’ schools that received a water-supply improvement, HP&WT and sanitation showed a reduction in diarrhoea incidence and days of illness.
18	Freeman et al. 2015 [23]	Kenya	Cross-sectional study	200 schools (20,000 children); in 70 schools, data collected on household and school WASH access and practice	Exposures: student shoe-wearing and soil-eating practices; household and school-level WASH conditions and access including latrine use, availability of drinking water, type and condition of latrines, availability of hand-washing facilities, soap for hand-washing, availability of tissue or water for use after defecation.	Soil-transmitted helminth infection.	Improved WASH access was generally, but not always, associated with lower intensity of STH infection. For school sanitation factors, the type of toilet, toilet conditions, and pupil to latrine ratio were all associated with overall or worm-specific infections. No clear trend of the relative importance of school versus household-level WASH emerged.
19	Garn et al. 2016 [24]	Kenya	Cluster-randomized trial	185 schools (divided into water-available and water-scarce groups)	School-level adherence to WASH interventions, as defined by the number of intervention components—water, latrines, soap—that had been adequately implemented.	Pupil diarrhea and soil-transmitted helminth infection.	There was reduced prevalence of diarrhea among pupils at water-scarce schools that adhered to 2–3 intervention components. In water-available schools, there was no evidence of reduced diarrhea with better adherence. No evidence of association between adherence and STH infection.
20	Garn et al. 2017 [44]	Mali	Matched-control trial	200 primary schools: 100 beneficiary schools and 100 matched control schools	Water and sanitation infrastructure, hand-washing facilities, wash supplies, hygiene promotion and capacity strengthening. Program fidelity (e.g., provision of water points and latrines) and program adherence (e.g., making soap available, maintaining latrine cleanliness) were also monitored.	Pupil diarrhea, respiratory symptoms, and absence from school.	Comprehensive WASH interventions that focus on adherence maximize the health effects of school WASH programs. WASH alone might not be sufficient to decrease pupils’ absenteeism.
21	Greene et al. 2012 [25]	Kenya	Cluster-randomized trial	135 public primary schools	Randomly assigned to: - Hygiene promotion and water treatment including closed buckets, taps, drinking water storage, water disinfection solution; teachers training on hygiene and behaviour change promotion. - As above plus ventilated improved pit (VIP) latrines. - Control group: receive intervention at conclusion of study.	*Escherichia coli* (*E. coli*) contamination on pupils’ hands.	Intervention did not reduce risk of *E. coli* presence on pupils’ hands; the addition of new latrines to intervention schools significantly increased *E. coli* risk among girls, with a non-significant increase among boys.
22	Grimes et al. 2017 [26]	Ethiopia	Longitudinal study	30 schools (3729 children provided blood, stool, and urine samples)	All schools were receiving school-feeding program from the United Nations World Food Programme. Half the schools received a WASH intervention upgrade.	School WASH infrastructure; student WASH knowledge, attitudes and practice; *Schistosoma mansoni*, *S. haematobium*, and soil-transmitted helminth infection; blood haemoglobin concentrations; height; and weight.	No statistically significant associations were found between home sanitation and hookworm. There were no reported findings on the added impact of the WinS intervention on students’ health.
23	Grover et al. 2018 [30]	Bangladesh	Cluster-randomised trial	20 government schools.	Allocated to one of four interventions: - Simultaneous handwashing infrastructure and nudge construction - Sequential infrastructure then nudge construction - Simultaneous infrastructure and high-intensity hygiene education (HE) - Sequential handwashing infrastructure and HE	Rates of handwashing with soap (HWWS) after a toileting event.	5 months post-intervention, the nudge intervention and HE intervention were equally effective at increasing HWWS after toileting. Simultaneous delivery of HE alongside handwashing infrastructure significantly outperformed sequential HE delivery; no significant difference was observed between sequential and simultaneous nudge intervention delivery.
24	Hetherington et al. 2017 [12]	Tanzania	Qualitative methods and pre- & post- questionnaire (participatory action research)	2 secondary boarding schools.	Train-the trainer model: Teacher workshops, school-based WASH lessons, extra-curricular activities, community events, “SHINE” clubs, non-stigmatizing activities to enable youth and communities to develop WASH strategies, a One Health sanitation science fair showcasing WASH projects.	WASH-related knowledge, attitudes and practices among students; level of engagement of students and community in the development and evaluation of sanitation and hygiene prototypes and health promotion strategies.	Statistically significant improvements in self-reported hygiene behaviour and knowledge, increased WASH communication. No changes in sanitation knowledge. Qualitative data highlighted WASH leadership among youth, enthusiasm from teachers and students, and community engagement.
25	Hunter et al. 2014 [38]	Cambodia	Quasi-experimental case-control longitudinal study	8 schools (4 case, 4 control)	Case schools received one 20 L container of treated drinking water per day (water treated by filtration and ultraviolet disinfection).	Weekly absenteeism rates.	A strong association between providing free safe drinking water and reduced absenteeism, though only in the dry season.
26	Karon et al. 2017 [13]	Indonesia	Cross-sectional study	75 schools (1780 students)	Beneficiary schools received: capacity building; improved toilet and water facilities and handwashing construction; hygiene promotion; strengthening of School Committees to create school WASH action plans.	The school hardware survey included questions about water, sanitation, hygiene, waste disposal and drainage. The student survey included questions on knowledge, attitudes and practice of hygiene habits at school and at home.	Intervention contributed to improved WASH infrastructure in schools, increased student communication with parents about hygiene, improved student WASH knowledge, increased rates of student handwashing after defecation, and lower reported rates of open defecation.
27	La Con et al. 2017 [31]	Kenya	Mixed-method cross-sectional study	28 schools	Handwashing and drinking water stations (containers with lids and taps on metal stands), bleach for water treatment, soap for handwashing, teacher-training, and educational materials.	Availability of soap and water at handwashing stations and treated drinking water 4 months after implementation; observation of student handwashing at stations both <10 m and >10 m from latrines; teacher-reported cleanliness and illness rates in pupils.	4 months after installation handwashing and water stations and education, pupils used handwashing stations in their schools and used stations located closer to latrines (<10 m) much more frequently.
28	Koopman 1978 [27]	Colombia	Cross-sectional	8219 school children (in grades 1–5) from 14 municipal schools and 17 private elementary schools	Exposure: classroom size and condition of school toilets (i.e., broken toilets, water on floor, used paper on floor, faeces in bowl, faeces outside bowl).	Prevalence of diarrhea, vomiting, common cold, and head lice.	Unhygienic toilet conditions, particularly faeces in the bowl, were related to increased diarrhea prevalence.
29	Migele et al. 2007 [28]	Kenya	Before and after study	1 private rural primary school (pilot project); 380 students.	Teachers provided education about behaviour change/safe water and hygiene. Schools were provided with water storage vessels and water tanks for handwashing; water was treated with bleach.	Student diarrhoea rates (assessed via review of local clinic records).	Findings suggest that diarrhea incidence rates decreased after implementation of the intervention.
30	Montgomery et al. 2012 [39]	Ghana	Non-randomized trial	120 schoolgirls aged between 12 and 18 years.	Three levels of treatment: provision of pads with puberty education; puberty education alone; or control (no pads or education).	School attendance.	After 3 and 5 months, pads with puberty education significantly increased attendance. Puberty education alone resulted in a similar attendance level.
31	O’Reilly et al. 2008 [14]	Kenya	Before and after survey	9 schools (with nine comparison schools for some indicators); 390 students. Final evaluation of 363 students and their parents	School-based safe water and hygiene programme: teachers trained on safe water system (SWS) and hand-washing; teachers instructed to form student safe water clubs, teach SWS and hygiene and encourage students to teach their parents. Schools provided with clay pots with narrow mouth, lid, and spigot; WaterGuard to treat water; water tanks with taps for hand-washing; soap.	School WASH facilities; stored water tested for chlorine. WASH knowledge and practices of students and their parents. Weekly absenteeism reports for 9 project schools and (for comparison) 9 neighbouring non-project schools.	The intervention reduced student absenteeism; safe water and hygiene knowledge transfer occurred from teacher to student; students’ knowledge of water treatment procedure increased significantly; students’ knowledge of appropriate times for hand-washing increased substantially; water treatment and hygiene knowledge transfer from student to parent and some evidence of behaviour change among parents.
32	Oster and Thornton 2009 [40]	Nepal	Randomized control trial	4 schools in rural Nepal, Chitwan province; 198 adolescent girls and their mothers.	Distribution of menstrual cups to adolescent girls in rural Nepal	School attendance and school-test scores.	No evidence that menstruation technology affects school attendance or test scores. Suggested that menstruation technology assists management of blood, but doesn’t reduce cramps and fatigue.
33	Patel et al. 2012 [15]	Kenya	Cluster randomized trial	42 rural primary schools	Safe water and hand hygiene education and installation of simple hand-washing and drinking water stations.	Student illness (respiratory illness and diarrhoea) and hygiene practices	The intervention produced improvement in hygiene knowledge and hand-washing techniques and a decrease in respiratory illness among students; no decrease in acute diarrhoea was observed.
34	Pickering et al. 2013 [16]	Kenya	Cluster randomized trial	6 primary schools in urban Nairobi (1364 students)	Schools randomly assigned to: - Teacher hygiene training plus provision of waterless hand sanitizer dispenser - Teacher hygiene training plus provision of liquid soap dispenser - Control: no intervention	Hand hygiene behaviour using structured observation; perceptions of soap and sanitizer (at follow-up).	Hand cleaning after toileting was 82% at sanitizer schools, 38% at soap schools, and 37% at control schools. Students at sanitizer schools were 23% less likely to have rhinorrhoea than control students (*p* = 0.02); reductions in self-reported gastrointestinal and respiratory illness were not statistically significant.
35	Saboori et al. 2013 [17]	Kenya	Cluster randomized trial	60 public primary schools	Regular provision of soap and latrine-leaning materials to primary schools.	Hand washing after latrine use and *E. coli* hand contamination among pupils	Observed hand washing with soap (HWWS) was significantly higher in schools that received soap (32%) and schools that received soap and latrine cleaning materials (38%) compared with controls (3%). There were no significant reductions in *E. coli* hand contamination.
36	Talaat et al. 2011 [41]	Egypt	Cluster randomized trial	60 elementary schools (30 intervention; 30 control)	Hand hygiene campaign: hand-washing twice per day in school; materials for students, teachers, parents; teacher’s guidebook for activities; hand-washing posters; student booklets; activities (e.g., theatres, song contests); campaign song; informational fliers for parents.	Laboratory-confirmed influenza A and B; student absenteeism and reasons for absence.	In the intervention group, absences caused by influenza-like illness, diarrhea, conjunctivitis and laboratory-confirmed influenza reduced by 40%, 30%, 67%, and 50%, respectively. The campaign was effective in reducing absenteeism.
37	Trinies et al. 2016 [18]	Mali	Match-control trial	200 schools (100 beneficiary schools; 100 matched comparison schools)	Installing/rehabilitating water points and latrines; distributing WASH supplies including soap, trash bins, disinfectant; hygiene promotion activities; training teachers, school management committees, school hygiene clubs; establishing financial, governance and management systems at the school level.	Recorded and self-reported student absence, and diarrhoea and respiratory infection among students.	There was a lower incidence of self-reported diarrhoea and respiratory infection among students in beneficiary schools. Students from intervention schools were less likely to report absence due to diarrhoea than pupils in control schools.
38	UNICEF 1994 [42]	Bangladesh	Cross-sectional	228 schools	Construction quality of water and sanitation system, rates of WASH infrastructure use, maintenance of WASH facilities, WASH knowledge among students, student hygiene behaviours.	Girls’ attendance rate at school	Girls’ school attendance rate was found to have increased following intervention.

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
