# Peer review of "Water, Sanitation and Hygiene (WASH) in Schools in Low-Income Countries: A Review of Evidence of Impact"

_ijerph, 2019, doi:10.3390/ijerph16030359_

Round 1
Reviewer 1 Report
This is a great piece of work; I appreciate the methodological soundness and the thoroughness with which the data (findings from peer-reviewed journal articles) is presented. The findings are relevant and do provide an important contribution.
What I am missing is some greater synthesis of the findings of the articles reviewed into a coherent argument. The paper does cluster the articles according to ‘outcome measures’, which is great. For improved clarity of the findings and argument of the article I would suggest the following amendments:
1. 28-32: These last 3 sentences could be merged into/ replaced by 1 sentence stating the key argument of the article.
2. Figure 1: nice!
3. 103-4: Why was no attempt made to evaluate the quality of studies or do a secondary analysis of study findings? I’m not suggesting that it would be necessary to do so; rather it would be better to provide some justification why it hasn’t been done.
4. 122: WinS – provide full phrase
5. 291-292: Table 2: this is a good schematic overview of literature, is very interesting, and does help with validity of research. Yet a large table like this interrupts the flow of the article. My suggestion would be to make it an appendix.
6. 293: The discussion section is perhaps the most important section of the paper, as here the findings are engaged with and put forward in a new light. The subjects of sections 3.2-3.6 are well placed in reporting on findings from the literature reviewed. For the discussion I would suggest an extension of these subjects, dedicating about a paragraph to each, not only to ‘intervention fidelity’ (308), which itself however is good! This would help the reader and argumentative clarity. An example of things that I would suggest taking up in the discussion and engaging with further are from section
- 3.2 (119…): WASH intervention in water scarce schools that receive water supply & sanitation improvements in addition to hygiene promotion and water treatment are more successful than schools that are ‘water-available’ or only receive hygiene promotion/ water treatment.
- 3.3: success of nudges in WASH interventions
The themes of ‘theory of change’, ‘management of menses’ are also nice here and relevant. (CLTS – might need to be written out?)
7. 329: 5. Conclusion:
Norm is to provide a brief summary of the sections in the paper, key findings/ themes discussed and how the research objective has been addressed.
336: The areas for further investigation listed here are good, could be emphasised a bit more.
343-354: paragraph style seems more suited for the introduction than conclusion. Perhaps cut it out?
Author Response
1: 28-32: The last 3 sentences have been merged into two sentences stating the key argument of the article.
2: Thanks for you comment on Figure 1.
3: 103-4: I now explain that as the studies use diverse methods and outcome measures no attempt was made to weight the value of findings according to study quality, or to conduct meta-analysis of study findings
4: 122 : WinS – provided full phrase
5: 291-292: Table 2 can be moved to the end, or as an appendix. I will leave this to the editorial team to make the move. If it is repositioned as an appendix I would need to change all references to 'Table 2' in main text.
6: I have developed a more extended discussion section that engages with the key findings in relation to disease, attendance, fidelity etc. As suggested, some sentences have been moved from the results sections and included in this discussion.
7. The conclusion has been revised as per suggestions. As suggested the paragraph including lines 343-354 (on SDGs) has now been moved to the discussion. I think it provides a global policy context to WinS and, while it could work in the intro it is probably better placed later on in the review so as not to distract from the focus.
Reviewer 2 Report
Dear Aurors,
thank you very much for the opportunity to review your article. I was pleased to read the very well structured and comprehensible article. The introduction provides sufficient information for an orientation on the topic. The methods and materials chapter is clearly described and graphically underlined with figure 1.
The selection and evaluation criteria are described in a comprehensible way.
Tables 1 and 2 provide a sound overview of the content and methodological orientation of the studies analysed. The results chapter is clearly structured and comprehensively and comprehensibly described. The numerous examples from the studies considered are helpful here..
In a few passages, aspects are listed that may fit better into the discussion section. an example of this is given in lines 223 et seq. where reference is made to further research needs to understand whether an how WASH interventions can improve habits and health outcomes among wider family members. ... may be such aspect coud be shiftened into the discussion section..
A further point is the more explicit discussion in the discussion section with the findings of the two review studies already mentioned in the introduction.
It would be appropriate to address this once again in the discussion section and to specify more explicitly what contribution the present study makes beyond that.
A further aspect that is worth emphasizing in the discussion concerns the influence of the methods used in the studies on the achievable statements and results - can conclusions be drawn from this for future research?
Regardless of this, this manuscript represents a valuable contribution that is worth publishing - taking into account the comments in the sense of minor revisions.
Author Response
Thankyou for your helpful comments.
I have carefully reviewed the findings and moved content that fits better into the discussion section (e.g. as suggested, I have moved the sentence focused on the value of further research on whether and how WASH interventions can improve habits and health outcomes among wider family members).
In the discussion section, I have included a short paragraph that makes clear how this review makes a contribution beyond the reviews on school-based WASH interventions (Jasper et al. 2012; Joshi and Amadi 2013).
The existing paper considers the methodological limitations of intervention studies. These have been highlighted further in the discussion section.
Reviewer 3 Report
I thank the authors for the opportunity to review this manuscript and believe it is an important piece of work in the WASH and larger public health literature. It is my hope that my comments aid the authors in improving the manuscript.
Major comments:
The paper is well-written as a whole, and addresses an important topic. My major comments are limited to:
Broadly, I believe the results section has most of the data it needs, but it could be structured in a format that more effectively conveys the information to the reader. As written, the information comes off a bit unorganized/haphazard and it does not do justice to the amount of work the authors put in. My main suggestion would be that the authors find a systematic way to present the information within sections so the overall evidence can be weighed more easily. This will also improve the authors’ ability to make more concrete points in the conclusions about where the gaps are. I understand that the authors did not intend to do a meta-analysis or grade the quality of the information, but some improvements in context and structure/order could make it clearer, specifically:
Ordering the sections by outcome seems appropriate, however, I would suggest that, within each section, the authors use a consistent order to present the information. This order could be on quality of study (e.g. synthesize results of RCTs first, then observational, before-and-after with/without control groups, etc.), or could be on technology (water interventions alone, hygiene alone, sanitation alone, water + sanitation, water + hygiene, etc.). Either way, presenting the information in these ‘categories’ in order each time will help the reader to group the studies by common themes (beside outcomes) and weigh studies against each other.
Regardless of the axis or variable chosen by the authors to order within each section, I would suggest including more information in-text about the type of intervention (e.g. explaining the facets of WinS interventions vs. water only) so the reader can better contextualize the differences. I realize much of this is presented in Table 2, but relevant syntheses at key points could aid in comparison of studies. The authors are correct that the scope of interventions is one of many important variables that could affect the ’effectiveness’ of the intervention.
Including reasons for the ‘failures’ of certain interventions seems to be important to the authors in many cases (e.g. the addition of commentary from authors of the primary literature about hypotheses as to why an intervention effect was/was not seen). This is fine to include, but would be clearer if also systematically described: e.g. instead of describing/mentioning for each study, characterize for the group of studies as a whole (X% of studies attributed failures to insufficient behavior change, etc.). It would be good to clarify, as well, which ‘explanations’ or ‘hypotheses’ are just that—explanations in the discussion with little data to back them up—vs. those that have documented data collected.
With regards to the ‘health’ section (3.2):
I’d suggest the authors use “diarrhea and WASH-related diseases”, “disease-related outcomes”, or some similar word choice rather than ‘health outcomes’-type diction (e.g. p.2, line 59). The WHO defines health not merely as the absence of disease and, as I default to myself as well, too often we assume health means diarrhea, etc.
In the section, it would be worth noting the potential for between-study variation in quality of the outcome measured. It looks like, it most cases, the outcome is self-reported diarrhea, but as Arnold et al. and others have shown, there are important differences in the quality of recall (as authors discuss in the discussion), which may make results of one study more/less useful than results of another. Importantly, did any studies use more objective outcomes, like stool specimen collection?
I would caution the choice of words/characterization of the study by Greene et al. 2012, mentioned in line 148 onwards. E. coli on hands is a hyper-variable outcome measure (see Pickering et al. 2011 and 2012 for some about this as well) that is not standard and is difficult to interpret in terms of disease-related outcomes. Presentation as ‘E. coli risk among girls’ (line 151) may be slightly misleading.
Within the results sections, I would suggest leading with the intervention fidelity section, as it makes the most sense in terms of the theory of change and is one of the most important points of the paper, in my opinion. The fidelity to an intervention should be quantified first and foremost before we can evaluate whether an intervention ‘worked’, so the authors putting it first in the results would emphasize that and help address some of the points above as well.
Minor comments:
Line 30: ‘yet delivering school-based WASH…’ sentence is a fragment.
Line 59: seems like ‘include’ should be ‘including’
Line 66-76: the authors separated by WASH intervention, but one assumes they also included multi-arm interventions? From the results, this seems clear, but they should mention this explicitly and perhaps present some statistics (like those in Table 1) for the types of interventions (W, S, H, WS, etc.).
Lines 183-187: I applaud the authors for this explanation of the intervention, and would suggest they include this type of brief in-text explanation for other studies when comparing them.
Line 208: ‘diarrhoeal’ should probably be ‘diarrhea’?
Author Response
Beyond the presentation of findings by sub-theme (i.e. key outcome measures), I have not developed additional critera for systematic presentation of the information within sections. Given the word limit, I have not attempted to summarise the findings of each study that is relevant to the five sub-themes within the narrative results section (i.e. disease-related outcomes, attendance, hygiene knowledge/behaviour, community/family changes, intervention fidelity); however findings have been identified for each study in Table 2.
I have removed reference to ‘health outcomes’ and replaced with terms such as 'disease-related outcomes'
There are a few additional sentences which summarise those studies that use objective measument of disease/infection (in the discussion section): e.g. stool samples for STH.
I now clarify that "it is important to note, however, that E. coli on hands is an outcome measure that is difficult to interpret in terms of disease risk and outcomes"
I have not moved the section on fidelity so that it leads the results section. Although this is a good idea I think the results section flows better by beginning with the more 'commonly considered outcomes of health, attendance etc. However, I have expanded the discussion on fidelity in the 'discussion' section.
Line 30: I have fixed the sentence that began ‘yet delivering school-based WASH…’
Line 59: ‘include’ has been changed to ‘including’
Line 208: ‘diarrhoeal’ has been changed to ‘diarrhea’